# An Interpretable Machine-Learning Algorithm to Predict Disordered Protein Phase Separation Based on Biophysical Interactions

**DOI:** 10.3390/biom12081131

**Published:** 2022-08-17

**Authors:** Hao Cai, Robert M. Vernon, Julie D. Forman-Kay

**Affiliations:** 1Molecular Medicine Program, Hospital for Sick Children, Toronto, ON M5G 0A4, Canada; 2Department of Biochemistry, University of Toronto, Toronto, ON M5S 1A8, Canada

**Keywords:** biomolecular condensates, machine learning, predictor, physical interactions, intrinsically disordered proteins, phase separation

## Abstract

Protein phase separation is increasingly understood to be an important mechanism of biological organization and biomaterial formation. Intrinsically disordered protein regions (IDRs) are often significant drivers of protein phase separation. A number of protein phase-separation-prediction algorithms are available, with many being specific for particular classes of proteins and others providing results that are not amenable to the interpretation of the contributing biophysical interactions. Here, we describe LLPhyScore, a new predictor of IDR-driven phase separation, based on a broad set of physical interactions or features. LLPhyScore uses sequence-based statistics from the RCSB PDB database of folded structures for these interactions, and is trained on a manually curated set of phase-separation-driving proteins with different negative training sets including the PDB and human proteome. Competitive training for a variety of physical chemical interactions shows the greatest contribution of solvent contacts, disorder, hydrogen bonds, pi–pi contacts, and kinked beta-structures to the score, with electrostatics, cation–pi contacts, and the absence of a helical secondary structure also contributing. LLPhyScore has strong phase-separation-prediction recall statistics and enables a breakdown of the contribution from each physical feature to a sequence’s phase-separation propensity, while recognizing the interdependence of many of these features. The tool should be a valuable resource for guiding experiments and providing hypotheses for protein function in normal and pathological states, as well as for understanding how specificity emerges in defining individual biomolecular condensates.

## 1. Introduction

Protein phase separation has recently been recognized as an important mechanism of compartmentalization in cells contributing to the formation of biomolecular condensates [1,2]. Liquid–liquid phase separation (LLPS) is not the only physical phenomenon that can contribute to the formation of these condensates, with these including sol-gel transitions and phase separation coupled to percolation (PSCP) [1,3,4]. Here we use the term “phase separation” as an imprecise shorthand for these mechanisms that rely on exchanging multi-valent interactions [5] that give rise to biomolecular condensates. Biomolecular condensates are found in a wide range of biological contexts, including intracellular condensates and membraneless organelles [6,7] such as signaling puncta [8,9], nuclear pores [10], transcription centers [11], and mRNA transport granules [12,13,14], as well as extracellular biological materials such as those in elastin [15,16,17], mussel foot [18,19], and squid beak [20,21,22,23]. Biomolecular condensates are also implicated in pathological aggregation (e.g., ALS [24] and Alzheimer’s disease [25]).

The physical mechanistic understanding of protein phase separation in all its complexity is challenged due to the richness and versatility of its driving forces. Phase separation can be affected by a large set of sequence-dependent factors, with a significant role of intrinsically disordered protein regions (IDRs) in many cases. For phase separation driven by IDRs, numerous weak interaction forces have been highlighted to contribute, including electrostatic interactions [26,27,28], pi–pi stacking [29,30,31], cation–pi interactions [19,26,32], and hydrogen bonding [33,34], with multiple forces often implicated as being seen in low-complexity aromatic-rich kinked segments (LARKS) [33], which exhibit kinked-beta-backbone hydrogen bonding and aromatic sidechain interactions. In elastin and elastin-like peptides, the hydrophobic effect is important for phase separation [35,36]. For phase separation driven by folded domains, specific sequence motifs, SLiMs [37], and their cognate folded binding domains are key; while these are an important driver of biological phase separation, our focus here is on IDR-driven phase separation.

Since most of the physicochemical factors that facilitate phase separation are sequence-dependent, there have been numerous efforts to use statistical learning to draw physical insights from known phase-separating sequences, i.e., to predict whether a sequence will undergo phase separation by comparing it against tested sequences, as previously summarized in a 2019 review [38]. However, the algorithms mentioned in that review focus on specific categories of condensates or biophysical features, and can only predict a subset of phase-separating proteins with high confidence. There is a high level of correlation among biophysical features, e.g., pi–pi and solvent interactions [29], electrostatic interactions and hydrophobic interactions [28,39], but none of these algorithms can estimate phase-separation propensities based on all of these physical forces, limiting the overall predictive capability of these “first-generation” predictors. In subsequent work [40], a machine-learning-based prediction tool (PSPredictor) that uses word2vec sequence encoding and the Gradient Boosting Decision Tree (GBDT) model outperformed all the “first-generation predictors” and achieved a 96% prediction accuracy. However, because of the design of word2vec encoding [41], its prediction results cannot provide quantitative information about the contributions from different driving forces, and therefore it lacks clear physical interpretability. Recently, a number of additional tools have been developed to quantify phase-separation propensity. One of these, PSPer, focuses on the prediction of prion-like RNA-binding proteins that phase separate using a Hidden Markov Model (HMM) [42]. PSPer showed good predictability (0.87 Spearman correlation score between its output and the critical concentration of FUS-like proteins); however, it has limited ability to predict phase-separating proteins that are not RNA-binding. Another, ParSe, combines two physical features, the hydrodynamic size of monomeric proteins and the beta-turn propensity estimated from polymer models, to predict phase-separation propensity; however, it only uses the composition and not the residue context when making predictions [43]. A third, PSAP, uses the compositional bias of phase-separating proteins and sequence-based biochemical features to train random-forest classifier with a 0.89 AUROC (area under the receiver operating characteristics curve), yet also lacks residue context in the prediction [44].

A major issue in developing a phase-separation predictor is the selection of a negative training set. Most recently developed predictors use sequences of the folded proteins in the RCSB Protein Data Bank (PDB) [45] as the negative set [29,40]; however, this leads to a bias towards a final classification algorithm that distinguishes between intrinsically disordered proteins/regions and folded proteins, since most proteins that are found to phase separate are IDPs or have IDRs. This classification does not identify the driving forces of phase separation, however, since many IDPs/IDRs are not phase-separating. In addition, many proteins phase separate during crystallization [46]. While most of these proteins do not contain IDRs and thus likely do not phase separate due to the sequence features of IDRs within their sequences, the PDB is not an optimal phase-separation-negative set for training a predictor of IDR phase separation. To avoid the issues with the PDB, in other cases the human proteome was chosen as the negative benchmark, bringing in higher structural complexity [33,47,48]. Another computational approach that has been developed to predict the propensity to phase separate, FuzDrop, has estimated that up to 40% of the human proteome can potentially undergo phase separation under certain conditions [49]. Therefore, it is clear that training a phase-separation-prediction algorithm on negative datasets such as the human proteome or PDB could include many false negatives, leading to significant challenges.

In the present work, we based our strategy on the idea that a combination of multiple different physical interactions drives phase separation, and developed a machine-learning-based predictor (LLPhyScore) that predicts based on a set of phase-separation-related physical interactions or features. While “LLPhyScore” was named by combining the acronym “LLPS” and “physical feature-based scoring”, the tool is not only focused on “liquid-liquid phase separation” but is intended as a general predictor of phase separation by various mechanisms that rely on exchanging multi-valent interactions within IDRs. We adapted the constrained training approach from our previous work on PScore [29] that focused on planar pi–pi interactions and extended it to a total of 16 (8 pairs) of physical measurements or features. The eight general features are not independent but are often discussed as separate terms. Our predictor development process was divided into two stages. In the first stage, we acquired sequence-based statistics (contact frequency/number of atoms/structure probability) from the PDB database of folded structures for each physical feature/interaction. We divided these observations by distinct residue pairs with varying sequence separation and developed a statistical method to predict the expected physical-feature values given a protein sequence. In the second stage, we trained the predictor to rank sequences by the weighted combination of the expected physical-feature values. During the predictor training, we used a genetic algorithm to optimize (i) the number of physical features to utilize in our final algorithm, (ii) the direction of contribution to the score (sign) of each feature, and (iii) the weights of each physical feature chosen for the final algorithm. The predictive model is a three-layer “neural network”-like model that infers the statistics of the input sequence based on physical features, residue types and residue counts and positions. The training revealed the better-appreciated importance of pi contacts and disorder, but also the less well-appreciated significance of solvent interactions, hydrogen bonds and kinked beta-structure. In order to address the “imperfect negative dataset” issue, we used three different negative training sets: the PDB, a curated human proteome, and a mixture of both the PDB and a curated human proteome, and examined their impact on the final predictor’s performance. The final predictor (LLPhyScore) achieved excellent predictive power (AUROC of 0.978) and demonstrated significant physical interpretability by providing a breakdown of the contribution from each physical feature to a sequence’s propensity to phase separate.

## 2. Methods

### 2.1. Data Preparation

The overall data preparation workflow is shown Figure 1 and includes the following four parts:

(1) The curation of phase-separation-positive (PS-positive) sequences: In this paper, we defined PS-positive proteins as proteins that can undergo phase separation on their own in vitro. We noticed that in several recently published phase-separation sequence databases, including LLPSDB [50], PhaSepDB [51] and PhaSePro [52], two main issues exist: (i) Many phase-separated systems are multi-component, comprised of “scaffold” proteins that are PS-positive and “client” proteins that are phase-separation-negative (PS-negative) on their own. However, “client” proteins were often mislabeled as PS-positive. (ii) There were many sequence errors (e.g., missing fluorescence tags; incorrect species; mishandled mutations and cleavages). To tackle these issues, we screened 142 papers (Appendix A) from July 2013 to January 2019, excluded sequences that can only undergo phase separation with DNA/RNA/other proteins from our positive set, and manually extracted 565 sequences (see Appendix A) as our PS-positive set (workflow shown in Figure 1). Then, we used LLPSDB and PhaSepPro to cross-check the sequences.

(2) The clustering of PS-positive sequences (for train–test split): A common practice used in other related work [29,40,48] is to split the training and test data in order to use previously discovered sequences for training and newly found sequences for testing. However, we noticed that there are many similar sequences reported at different times (e.g., sequences from a family that was worked on by the same lab in many years, or different mutants of the same wild type), so performing a time-based split will cause greater bias as we are training and testing similar samples on the algorithm. This issue would be even more problematic considering the limited sample size for PS-positive proteins reported at the start of our work (<1000 samples). Therefore, before splitting the training and test set, we applied a hierarchical clustering to 565 PS-positive sequences, and obtained 157 sequence groups, as shown in Appendix A, where sequences within the same group have a pairwise similarity of higher than 50%. The subsequent train–test split was then conducted based on sequence groups instead of individual sequences, so that training and testing set proteins are derived from separate sequence groups.

(3) PS-negative training sets: We created two negative sequence databases: (i) the PDB sequence database, from which we collected 16,794 sequences (see Appendix A) from high-resolution (≤2.0 A) structures in the PDB, and (ii) the curated human proteome sequence database, from which we collected 20,380 human proteome sequences (see Appendix A) from Uniprot and removed sequences with either null values or high values (top 20%) in CRAPome [38,53]. We chose CRAPome as the method of filtering out phase-separation-prone sequences because it is an empirical measurement, rather than a prediction, of non-specific interactions in human proteins [53]. This resulted in a “clean” human set of 6102 sequences (Appendix A). Appendix A contains the CRAPome (along with final LLPhyScore) scores for all of the human sequences, including both those within the curated negative training set and those not in the curated list. It is worth noting that false negatives existed in both the PDB and curated human negative sequence data sets. While we attempted to minimize false negatives, both the PDB and curated human sets were compromised to an unknown degree. Certainly, there were fewer positives in these sets than in the known PS-positive sequences, but perfect discrimination is likely impossible because the training sets are not gold-standard truths, and the percentage of human-proteome- and PDB-derived sequences that undergo phase separation is unknown.

Since the positive sample size was much smaller than the negative sample size, we then randomly selected 3406 sequences each from the (i) PDB sequence database and (ii) curated human sequence database, and constructed two negative sets: (a) the PDB set (3406 sequences) and (b) the curated human set (3406 sequences). Finally, we mixed (a) and (b) at a 1:1 ratio and built (c), a mixture of the PDB and the curated human set of 3406 sequences (randomly selecting 1703 from PDB and 1703 from human). During the initial predictor training, the PDB set was used as the main set for determining both the “signs” of the features and the number of features to retain; then, all three sets were used to optimize the “weights” of the features and to compare the three final predictors’ performances with each other and other predictors.

(4) The construction of the training/test/evaluation sets: For the training of the predictor models and the optimization of the model parameters, we initially constructed training and test sets by adopting a 70–30% train–test split ratio for the PS-positive and negative sample sets in steps (1) to (3). For the positive samples, random sampling was conducted at the clustered group level until >30% of sequences went to the test set. However, due to the existence of large sequence groups (30–50 sequences), the end result was actually close to an even ratio with 305 sequences in the training set and 260 in the test set, as shown in Figure 1. Therefore, we then used an even ratio between the training and test sets for the negative samples, where the random split was conducted at the sequence level, given that the issue with similar sequences did not affect the negative set.

For the evaluation of the final models’ performances trained on different negative databases (PDB, human, human + PDB) on a defined dataset, we constructed evaluation set 1, composed of the entire PS-positive set (565 sequences) and the entire PDB proteome (16,794); for the comparison of our predictor’s performance with other state-of-the-art phase-separation-prediction algorithms, we constructed evaluation set 2, composed of the entire PS-positive set (565 sequences) and the entire human proteome (20,380 sequences).

For more details on the constructed training, test and evaluation sets, see Appendix A.

### 2.2. Construction of Physical-Feature Collection in LLPhyScore

We made two core assumptions in this work to develop a sequence-based predictive algorithm: (i) phase separation is driven by multiple physical forces and structural factors; (ii) for any phase-separated system, these forces and features together build up the system’s free energy to drive the phase transition. Then, we constructed a set of 16 (8 pairs) sequence-based, phase-separation-related physical features, including weak interactions and structural patterns, as described below. More details on each of these can be found in the Technical Methods and Appendix A. The motivation for our design derives from the focus of much of the phase-separation literature on protein–protein interactions, often ignoring protein–water interactions (see below), and the assumptions that one or a few certain specific physical or chemical interactions are dominant contributors or that some are not important (e.g., h-bonding, kinked beta). We also initially hypothesized that proteins found in distinct biomolecular condensates would use specific types of physical “interactions” based on our definitions as a way to generate specific condensates.

Protein–water interactions. As pointed out by others in the field [36,54], protein–water interactions represent a largely overlooked driving force in phase separation because of its synergistic nature with other interactions such as pi–pi, hydrogen-bonding and electrostatic interactions [29,36,54]. Here, we considered it as a separate force/feature and explored its role in phase separation. We defined protein–water interaction by contacts, and measured solvation contacts and hydrophobic contacts using two inversely correlated terms, a residue–water interaction count and a residue–carbon interaction count. The frequency measurement followed the same protocol as for pi–pi interactions in PScore [29].

Helices and strands. While most phase-separating proteins contain IDRs that play a significant role in driving phase separation, in some cases [33,37,55], these IDRs transiently exhibit a folded structure (either helices or beta-structures with varied dynamics and sizes) that can play a critical role. Here, we used the DSSP program to assign the secondary structure [56] and enable helices (H) and strands (E) to be considered as contributing features. Disorder was categorized as a separate feature, because most reported phase-separating systems are IDR-driven, and the statistics are highly skewed towards disorder, which could be detrimental to the algorithm training. Boolean values (true or false) instead of frequency were utilized for helices and strands.

“Long-range” and “short-range” disorder. Due to the large difference in structural context between short (<5 residues long) and long (>15 residues long) disordered regions [57,58], disorder was divided into these two categories. Here we defined the presence or absence of disorder as Boolean values (true or false), and measured disorder based on the lack of helix or strand DSSP assignment of consecutive residues in a sequence.

Long-range and short-range electrostatic interactions. Electrostatic interactions have been established as another important driving force for phase separation, especially for highly charged sequences in complex coacervation systems, such as for the tau protein [59]. Here we defined electrostatic interaction using coulombic interaction energy with atomic partial charges taken from the Talaris2014 force field [60], dividing the interaction energies by the sequence separation of the involved atom pairs into short-range (<5 residues apart) and long-range (≥5 residues apart). We note that complex coacervation will not be predicted as the approach is based on the phase separation of a single protein.

Long-range and short-range hydrogen bonds. Hydrogen bonding was found in some cases to co-exist with other driving forces, including pi–pi contacts [21] and protein–solvent interactions [61]. In this work, we considered it as a separate force and explored its role in phase separation. We used the PHENIX software suite [62] to identify OH-N hydrogen bonds and measured inter-residue hydrogen-bond interaction counts in short-range (<5 residues apart) and long-range (≥5 residues apart) contexts.

Long-range and short-range pi–pi interactions. We utilized our previous approach from the PScore phase-separation predictor based on planar pi–pi contacts [29], determining the contact frequency for residue pairs in the context of short-range (<5 residues apart) and long-range (≥5 residues apart) interactions.

Long-range and short-range cation–pi interactions. Cation–pi interactions were found to have a specific residue-type preference among the cations arginine and lysine and the aromatics phenylalanine, tyrosine and histidine, and the substitution of preferred residues in certain systems cause drastic change in phase-separation behavior [63]. In order to crudely estimate the potential cation–pi interactions, we adapted the electrostatic potential by adding partial negative charges above and below the planes of aromatic ring systems, balanced it with an in-plane positive charge, and then calculated the change relative to our standard electrostatic term. These measurements were again split into short-range (<5 residues apart) and long-range (≥5 residues apart).

Kinked beta-strands (K-Beta). It has been observed that specific sequences from some phase-separating proteins can form fibrils of kinked beta-strands, with beta-strand hydrogen bonding occurring without extended backbone torsion angles and forming fibrils similar to amyloids [33]. The prediction of this feature has previously been performed by the energetic assessment of a sequence’s ability to adopt the topology found in these fibril structures [64], and we created an analogous classification strategy by identifying sequences in the PDB that were similar or dissimilar (measured by backbone RMSD) to these kinked beta-strands [65]. Two Boolean metrics, K-Beta similarity and K-Beta non-similarity, were determined from RMSD values after the structural superposition calculations.

Based on the above 16 (8 pairs) features, we designed a sequence-representation system (See Technical Methods and Figure 2) to convert a sequence into inferred residue-level feature values (frequencies/numbers/Booleans). Note that many of these features are highly interdependent, particularly protein–water interactions with all of the others, cation–pi with pi–pi and electrostatics, and kinked beta with hydrogen bonds and pi–pi [33]. In addition, the role of residue-type preferences, which are also terms that are fit during training (including counts and positions), cannot easily be deconvoluted from these features.

## 3. Results and Discussion

### 3.1. Predictor Training

The concept of “predictor training” in this work means: (i) for a specific sequence, the algorithm outputs a summed score calculated by a weighted combination of the expected physical-feature values, and (ii) during the predictor training, we optimized the combination of physical features, as well as the “weight” for each feature. The workflow of the predictor training is shown in Figure 3.

The predictor training has three outcomes, described here:

(1) “Signs” of features were determined using individual feature training. Some features in our list were positively correlated with the performance of the developing predictor, while other features were negatively correlated. Therefore, before combining the 16 features, we first trained each feature individually and let the algorithm decide the “direction” (positive or negative) of its correlation with performance (measured by AUROC). As shown in Figure 4 and Appendix A, the features that were found to correlate negatively were protein–carbon interactions, the helical secondary structure, long-range electrostatic interactions, both short- and long-range cation–pi interactions and the kinked-beta (K-Beta) non-similarity. While the negative correlation for protein–carbon interactions and K-Beta non-similarity are consistent with an understanding that these features do not contribute to phase separation, in general these results are not simply interpretable as contributing positively or negatively to phase separation. This is particularly the case for electrostatic interactions including cation–pi, as it is not clear how the predictor deals with locally repulsive electrostatic interactions (clustered charges) that may favorably interact over longer ranges with oppositely charged clusters, or how well our crude estimate of cation–pi interactions works. Certainly, complex coacervation was not predicted as this tool was limited to homotypic phase separation, i.e., involving a single protein sequence.

(2). The number of features to include was determined using competitive feature training. After determining the “signs” of the features and applying them, we combined 16 features and allowed them to “compete” with each other through “competitive” training, then ranked their importance based on the final contribution (positive or negative) of each feature to distinguishing phase-separating from non-phase-separating proteins, as shown in Figure 5. While all 16 features achieved an average z-score greater than 1.5, the average z-scores for protein–water, protein–carbon, long-range hydrogen-bond and long-range pi–pi interactions were larger than 3.0, and those for disorder (within both short and long segments) and kinked-beta similarity were larger than 2.5. While the competitive training approach suggests the ability to quantitatively compare the significance of these physical interactions in phase separation over the input positive set, the interdependence of the terms and the convolution with the residue-type preference makes this comparison much more qualitative. We then came up with three different combinations of features according to the ranking, combining the top 8 or top 12 features based on ranking or combining all 16 features, in order to identify the minimal number of features that provides both good performance and physical interpretability. We conducted competitive training on each of the 8-, 12-, and 16-feature algorithms and assessed their performance. As shown in Appendix A, the combinations of 12 and 16 features did not demonstrate better performance than the combination of 8 features. To avoid overtraining, we chose the 8-feature combination in the final predictor training, with the weights of the smaller number of terms from training (see “(3)” below) reflecting the contributions of the features that were dropped. Thus, the choice of eight features cannot be interpreted as these features being the only ones that physically contribute to phase separation or that the resulting predictor ignores the contribution of those features. Cation–pi interactions are a clear example of this, as they are represented in the 8-feature predictor as a combination of residue-type preference, electrostatics and pi–pi interactions, even though they are not discretely represented as their own term.

(3) The “weights” of features in the final predictor were determined using competitive feature training on the entire dataset. We built the final predictor (“LLPhyScore”) and optimized the “weights” for the chosen eight features with their respective signs by competitive training on training set 1 and tested the model performance on test set 1 (Appendix A). We chose AUROC as the model-performance metric, which was 0.969 for training and 0.942 for the test (Appendix A) with the PDB as the negative set. This indicates that minimal overtraining occurred during the “weights” optimization. Then, we trained the “weights” again on training set 1 + test set 1 to yield the final predictor called “LLPhyScore-PDB model” based on its use of the PDB as a negative set. The LLPhyScore–PDB model achieved an AUROC value of 0.978 (Appendix A) on evaluation set 1 (including all PS-positive sequences and the full PDB proteome, Appendix A) and good separation between positives and negatives (Appendix A).

### 3.2. Model Performance Comparison against Different Negative Training Sets

As noted previously, there is no perfect negative sample set for phase-separation-predictor development; therefore, after we trained the LLPhyScore-PDB model (on training set 1 + test set 1), we also trained the LLPhyScore-Human model (on training set 2 + test set 2) and the LLPhyScore-Human + PDB model (on training set 3 + test set 3), and evaluated the three final models using both evaluation set 1 (all PS-positive sequences and full PDB proteome) and evaluation set 2 (all PS-positive sequences and full human proteome). The results shown in Figure 6 and Appendix A indicate that the PDB model showed the best performance on evaluation set 1 against the PDB (AUROC of 0.978), but the worst performance on evaluation set 2 against the human proteome (AUROC of 0.824); the human model showed the best performance on evaluation set 2 (AUROC of 0.941) and the worst performance on evaluation set 1 (AUROC of 0.908). This indicates that the negative training set of different models had a significant impact on the final model performance. The model using only folded proteins from the PDB as the negative training sequences tended to have less power to generalize on evaluation set 2 (including the full human proteome), which contained many disordered regions. On the other hand, the model only using human proteins as the negative training sequences still had a strong ability to discriminate most PS-positive sequences from PDB sequences in evaluation set 1. This is also reflected by the fact that the human + PDB model showed a more balanced result for evaluation set 1 (AUROC of 0.947) and evaluation set 2 (AUROC of 0.933). Together, these results support the use of the curated human proteome as a negative set, alone or with the PDB, and our choice of the human + PDB model as the optimal model.

### 3.3. Predictor Validation

To validate the final predictors’ performances, we constructed three sets of baselines. (1) Instead of providing PDB-based physical-feature values to the genetic algorithm, we provided random values from a normal distribution N(0, 1) in the weight-training step. (2) Instead of providing sequence-based physical-feature values, we provided random values from the distribution of residue-specific physical-feature values. (3) Instead of optimizing 20 weights for 20 residue types for each physical feature, we optimized 1 weight for all 20 residue types for each physical feature (removing residue specificity) during training. As shown in Figure 7 for the human + PDB model and Appendix A for all three models, baselines 1 and 2 showed a very high training AUROC but a low test AUROC, whereas the final models had both high training and test AUROCs. This suggests that the final predictors’ good performances did not result from overtraining the genetic algorithm, which was the case for baselines 1 and 2. The comparison between baseline 3 and the final models also suggests that it is important to have residue specificity in our model for good prediction performance.

### 3.4. Comparison of Prediction Using Eight Features or Single Features

To test whether a combination of eight features can outperform the prediction using a single feature, we extracted from the three final models each of the feature components as one-feature predictors and evaluated these one-feature predictors on evaluation set 1. As shown in Figure 8a and Appendix A, the receiver operating curves (ROCs) of one-feature predictors were outperformed by the eight-feature predictors. We also plotted Venn diagrams showing their recalled sequences at a confidence threshold that returns 2% of the PDB as a positive result (chosen based on the methods described in previous work [32,38,40]) as shown in Figure 8b and Appendix A. We observed that each of the one-feature predictors missed a number of sequences (48–350 sequences) that were captured by the eight-feature models. This result supports our underlying assumption that phase separation is driven by a combination of different physical features, and that driving forces for different sequences can vary.

### 3.5. Comparison between LLPhyScore and Other Phase-Separation Predictors

We compared the performance of our predictor (LLPhyScore, three final models) with PSPredictor, as well as two first-generation predictors, PScore and catGRANULE, in Figure 9. The comparison was conducted on both evaluation set 1 (PS-positive sequences and the entire PDB proteome) and evaluation set 2 (PS-positive sequences and the entire human proteome).

We can see that the LLPhyScore-PDB model showed the best performance on evaluation set 1 and even slightly outperformed PSPredictor, which was trained against 5258 sequences from the PDB. The LLPhyScore-PDB model also showed a better AUROC than PScore, which is based solely on planar pi–pi interactions. The LLPhyScore-Human + PDB model showed a slightly decreased performance on evaluation set 1 compared to the LLPhyScore-PDB model; however, it was still better than all of the first-generation predictors. The LLPhyScore-Human model showed a comparable performance to PLAAC.

However, on evaluation set 2, the LLPhyScore-PDB model did not show better recall statistics than the other first-generation predictors until a 30% acceptance threshold, as shown in Figure 9b. This is in line with the estimate of up to 40% of the human proteome driving phase separation [49], and could be considered support for an estimate of at least 30% of the proteome being involved in phase separation. On the other hand, the LLPhyScore-Human model and LLPhyScore-Human + PDB model both showed good performance on evaluation set 2, indicating that, by mixing the human and PDB sequences, the training algorithm can optimize PS-positive sequences from both negative sets. We also see (Figure 9a,b) that the LLPhyScore-PDB model showed comparable recall trends with FuzDrop. As a phase-separation predictor also based on biophysical principles combined with statistical training, FuzDrop uses a protein’s binding entropy as the target function. The fact that the LLPhyScore-PDB model and FuzDrop showed similar statistics supports the utility of approaches directly addressing the biophysical features and energetic driving forces underlying the formation of condensates.

### 3.6. Feature-Based Breakdown of Scores for Different Sequences

To further explore the general expectation that the phase separation of different sequences can be driven by different physical features, we clustered PS-positive sequences based on their single-feature scores after normalization. As shown in Figure 10 and Appendix A, FUS, Nup98, an elastin-like peptide (ELP), and MEG-3 were categorized into different clusters, which demonstrates the ability of LLPhyScore to treat different types of sequences, although most proteins were not clearly distinguishable. This underscores the interdependence of many of the physical features. For the LLPhyScore-Human + PDB model, the breakdown of the scores (Figure 10) shows that Nup98 has high scores for protein–carbon interactions but low scores for disorder, pi–pi interactions, and K-beta, whereas for FUS, the scores are high for most of the features.

### 3.7. Gene Ontology Term Enrichment

To explore our hypothesis that different biomolecular condensates would include proteins driven by similar features, we analyzed the enrichment of GO terms for human proteins in the top 10% (high confidence threshold) of scores from the LLPhyScore models predicted by eight single features as well as the combination of eight features in the final predictors. As shown in Figure 11, Appendix A, most GO terms identified by first-generation predictors [38] and by PSPredictor [40] were also enriched in sequences identified by LLPhyScore, such as extracellular matrix and nuclear body. For certain annotations associated with phase separation such as cytoplasmic stress granule, postsynaptic density and transcription factor complex, we observed differences depending on which features were utilized, which suggests that, for different biomolecular condensates with different functional roles for phase separation, the features linked to phase separation are also different, and are rooted in their sequence-specific biophysical landscape.

### 3.8. Physical Insights into Phase Separation Based on LLPhyScores of the PDB Set

The assessment of the physical basis of the LLPhyScore predictions is complicated not only by the interdependence of the features but also by the detailed choices made during the training process, where the weight given to a feature by the final model not only reflects that feature but the full sequence context of the residue including residue-type preferences. Therefore, to assess how scores relate to the physical features for which we trained, we applied the predictor to the sequences of known structures in order to assess phase-separation scores by directly comparing them to “true” measurements of sequences in observed structural contexts. For this, we scored each amino acid independently, comparing the physical features associated with being in the top 50% of scores against the overall distribution for that residue type. Figure 12 shows high score enrichment statistics for a variety of physical features, including secondary structure (a), short-range pi–pi interactions (b), kinked-beta similarity and dissimilarity (c), disorder (d), short- and long-range electrostatics (d,e), and local water/carbon contacts (f,g).

For the protein sequences found in our PDB set, the predictor generally assigned low scores to structures that can satisfy their interactions locally. Helical residues that fully satisfy their backbone hydrogen bonds typically had low scores (Figure 12a), as did residues with stabilizing charge interactions found between nearby local residues (Figure 12e). Notably, charge interactions between non-local residues (Figure 12f) had above-average scores, consistent with the known effects of blocks of like charges in driving phase separation [26,66,67]. For short-range electrostatics (Figure 12e), attractive interactions (negative numbers) were not favorable and repulsive interactions (positive numbers) were, with long-range electrostatics (Figure 12f) generally flipping this relationship, which is consistent with the idea of locally self-satisfied interactions not being favorable.

For secondary structure, there appeared to be three categories of effects based on backbone hydrogen-bonding satisfaction and torsion-angle regularity. Fully self-satisfied structures, specifically helices, had the lowest scores. Ordered but not necessarily locally-satisfied structures, which include beta-strands as well as 3_10_ helices (often associated with short helices [68,69] and capping motifs [70]), had intermediate scores. Irregular secondary structures, including elements with defined hydrogen-bonding patterns (turns, bulges, and bent/kinked strands), as well as solvent-bound loops, had the highest scores. In general, the ability to form hydrogen bonds with a solvent was consistently associated with higher scores, as was the lack of a repetitive ordered structure. In this analysis, bent and twisted strands typically scored better than fully disordered residues, especially for proline, suggesting that the availability of backbone hydrogen bonding plays a role, and not just the lack of structure.

The differences between disorder prediction and phase-separation prediction are further defined in Figure 12d. In general, disordered residues were more likely to score high, with long stretches of disorder scoring higher than short disordered loops. However, the majority of this bias results from hydrophobic or aromatic residues, specifically V, L, I, F, H, Y, and W. This is consistent with disorder on its own being insufficient for phase separation, with disorder that forces hydrophobic and aromatic residues into contact with the solvent supporting phase separation.

This indirect solvent relationship can also be directly observed by the measurement of solvent interactions and overall burial, as shown in Figure 12g,h. In general, residues with a high number of observed water contacts had higher scores, and residues with a high degree of burial (assessed by the number of carbon contacts) had lower scores. However, this trend was more pronounced for hydrophobic residues and was not observed for polar or negative residues (N, Q, E and D). This may be expected given that the hydrophobic effect is driven by the solvent, with the energy associated with a reduction in solvation driving hydrophobic residues together (i.e., solvent relationships are what makes hydrophobics sticky). In this context, we observed that hydrophobic residues that were forced to be in contact with the solvent by their local sequence context were predicted to contribute to phase separation.

The notion that sequences that force solvation are prone to phase separation matches the observations for the secondary structure. We note that while extended beta-sheets can often exclude solvent, by forming flat planar interactions with other sheets, kinked beta-strands cannot. Figure 12c shows that sequences with high structural similarity to kinked beta-structures had higher scores, especially for hydrophobics and aromatics.

Together, our analyses of the LLPhyScores for the PDB structures supports the view that disorder itself does not drive phase separation, but locally unsatisfied sequences that are constrained in their ability to exclude the solvent, including those that can adopt an irregular or kinked beta-structure to contribute backbone hydrogen bonds, do drive phase separation. These results may contribute to the current discussion of the role of sequences with the propensity to form a kinked beta-structure in protein phase separation [33,64].

### 3.9. High-Scoring Structures in the PDB Trend towards Disorder

The protein structure databank is often used as a negative set when training phase-separation classifiers, but this is not a ground truth, and the true fraction of proteins with structures present in the PDB that are also capable of driving phase separation is unknown. To demonstrate this issue, we scored the set of PDB reference sequences used in this study and observed that the highest-scoring proteins were not random; the score selected for proteins with significant disorder relative to the average structure found within the PDB (Figure 13). This was true using multiple definitions of disorder: (i) of the highest-scoring 1% (N = 167), 99 had more than 50% of the reference sequence missing from the density (Figure 13a), and (ii) for the residues that were found within the density (Figure 13b), 128 of these proteins had more than 50% of the residues in secondary-structure classes other than helix and strand, with 62 of these having more than 50% of their residues in contiguous loop/turn/random coil segments of four or more residues in length.

The highest-scoring sequences for LLPhyScore in the PDB depart significantly from the expectation of well-ordered folded domains, and their function is unlikely to be defined simply by their ability to form the state observed within these structures. These results, in addition to describing physical features that are correlated with phase separation, highlight the need for a biophysically defined empirical negative set for future work in training phase-separation classifiers.

## 4. Conclusions

In this work, we demonstrated the utility of combining different physicochemical interactions as driving forces in the prediction of protein phase separation. We addressed the issue of the “imperfect negative training set” by training three predictor models on three different negative sets and compared their performances. We optimized the combination of physical features in the final predictor models and achieved a superior performance over first-generation predictors. Importantly, our predictors enable a physical interpretability that is not possible with another comprehensive predictor, PSPredictor. Our results are consistent with the understanding that phase separation is driven by a combination of inter-related physical factors, including protein–water interactions, pi–pi contacts, disorder, hydrogen bonding—such as in the context of a kinked beta-structure—and electrostatics. By clustering sequences based on their physical-feature scores, we can differentiate some phase-separating sequences by their contributing driving forces, suggesting one contributor to the basis for specificity in the formation of the large number of unique biomolecular condensates found in biology. However, we found that many proteins used combinations of most or all of the features, reflecting their highly interdependent nature. We also observed that almost all the features were correlated with protein–water interactions. Therefore, the idea of protein–protein interactions driving phase separation themselves is simplistic, and for biomolecular condensates there is likely always a three-way interaction involving two or more protein groups and water. LLPhyScore should be a useful tool for the protein phase-separation field to provide hypotheses regarding key interactions driving phase separation, as well as for screening proteins that may play important biological roles in the context of biomolecular condensates.

## 5. Technical Methods

### 5.1. Curation of PS-Positive Sequences

We performed a search on PubMed for all papers published from July 2013 to January 2019 that contained keywords related to phase separation (“phase separation”, “liquid condensates”, “membraneless organelles”, etc.), and manually screened 142 papers out of 689 articles that described in vitro phase-separating systems. Then, we extracted all the sequences from the papers (main content/supplementary information/Uniprot entry) that had clear evidence of phase separation on their own (either a detailed phase diagram, or mentioned as “phase separation positive” in the text) in the content. A total of 565 sequences were extracted and were checked twice (Appendix A).

### 5.2. Clustering of PS-Positive Sequences

The clustering of positive sequences was performed by hierarchical clustering (shown in Appendix A). First, a 20 × 20 dipeptide count number grid was calculated for each sequence, with each number being the number of a residue pair (e.g., AG) in the sequence (Equation (1)). Then, a Jaccard similarity value was calculated for any two sequences by dividing the overlap of the union of two 20 × 20 grids (Equation (2)). If two sequences had different lengths, then a sliding window of the smaller length was applied to the longer sequence, and the highest similarity value calculated for all sliding windows was kept. Finally, we used the hierarchical clustering package in Python scikit-learn [71] to conduct the clustering for 565 sequences. A cutoff similarity threshold of 0.5 was chosen.
(1)A,B=[N(Ala−Ala)⋯N(Ala−Val)⋮⋱⋮N(Val−Ala)⋯N(Val−Val)]=(aij)∈ℝ20×20,(bij)∈ℝ20×20
(2)J(A,B)=|A ∩ B||A ∪ B|=∑i=120∑j=120min(aij,bij)/∑i=120∑j=120max(aij,bij)

### 5.3. Preparation of PS-Negative Sequences

Two PS-negative sequence databases were prepared. First, 16,794 sequences were collected from high-resolution (≤2.0 A) structures in the PDB as the first negative sequence database (“PDB base”; Appendix A); Second, 20,380 human proteome sequences were collected from Uniprot [72] (Appendix A), then we used their CRAPome scores calculated in Vernon et al. [38] as a filter for PS-positive sequences. Sequences with either null values or high values (top 20%) in CRAPome were removed, resulting in a “clean” human set of 6102 sequences (“Human base”; Appendix A). The CRAPome-filtered curated human proteome set should have fewer positives than the uncurated human proteome, with final LLPhyScores shown for these two sets in Appendix A, demonstrating an overall shift to negative scores for the curated sequences.

From these two PS-negative sequence databases, three negative sets were prepared: (a) a PDB set, including 3406 sequences randomly selected from the PDB base; (b) a human set, including 3406 sequences randomly selected from the human base; (c) a human + PDB set, including 1703 sequences randomly selected from the PDB set and 1703 sequences randomly selected from the human set.

### 5.4. Construction of Training/Test/Evaluation Datasets

The construction of the training set and test set began with PS-positive sequences. Random sampling was conducted on 565 PS-positive sequences at the clustered group level, with 305 sequences assigned to the training set and 260 sequences assigned to the test set. Then, a 50–50% split ratio was applied to three PS-negative sets at the sequence level, with 1703 sequences from each set assigned to the training set, and another 1703 sequences assigned to the test set. A total of three training–test set pairs were constructed accordingly.

Two evaluation sets were constructed. (1) The entire PS-positive set (565 sequences) + the entire PDB base (16,794); (2) The entire PS-positive set (565 sequences) + the entire human base (20,380 sequences).

For more details, see Appendix A.

### 5.5. Physical-Feature-Based Sequence Representation

Eight different pairs of general phase-separation-driving factors were defined to represent a protein sequence, resulting in a total of 16 physical features, as summarized in Appendix A. For each of these features, its sequence-based statistics (contact frequency/number of atoms/structure probability) in the PDB were acquired by mining the structures of folded proteins in the PDB. The observations were split by distinct residue pairs with varying sequence separations, leading to a database of “feature values”, with each “feature value” being an empirical, per amino acid energy potential corresponding to the frequencies of specific contact types in the PDB. Then, for a given input sequence, inferred “feature values” for each residue of this sequence were obtained by matching its residue pair and sequence context to the “feature value database”. For example, the short-range pi–pi contact frequency for valine in the tripeptide valine–glycine–tryptophan can be inferred by taking the average short-range pi–pi contact frequency for the residue pair valine–glycine with 0 separation and valine–tryptophan with 1-residue separation (see also Figure 2).

Specific definitions for each of these are as follows:

Pi–pi Contacts. Pi–pi contacts were defined using the method in Vernon et al. [29], and then divided into short-range and long-range by sequence separation. Less than five residues apart was defined as short-range, and greater than or equal to 5 residues apart was defined as long-range.

Hydrogen Bonding Terms. Structures were probed for OH-N hydrogen bonds using PHENIX [62], with the following commands used to extract hydrogen-bond information.

Phenix.reduce -Quiet -FLIP [pdb file] > /PHENIX_ALL/PHENIXL.pdb

Phenix.probe “NITROGEN,OXYGEN,HYDROGEN” -Quiet -ONEDOTeach -NOCLASHOUT -SUMMARY -NOVDWOUT. /PHENIX_ALL/PHENIXL.pdb|grep greentint > /N17.PHENIX/HLIST.txt

Bonds were than classified as short-range and long-range by sequence separation (short-range < 5, long-range ≥ 5).

Water/Carbon Contact Counts. Water and carbon counts were calculated only for the subset of proteins in our training set that had a total number of water molecules greater than the number of protein residues. This captured almost all of the models with a resolution ≤ 1.8 but removed lower-resolution models. Counts were measured for residues in their crystallographic context (measurement includes atoms from symmetry partners).

Secondary Structure. The DSSP letter code was used for secondary-structure assignments, with H/G used for helix, E for strand, and all others binned to loop.

Disorder. For identifying disordered residues, a DSSP assignment of “not G/H/E” over a span of at least 3 residues was used to classify residues as loops. These loop residues were then assigned as short disorder if they fell within 3 residues of G/H/E and as long disorder if they did not.

Charge. PHENIX (via the phenix.reduce command) was used to complete the PDB structures by adding hydrogen atoms, and charge interactions were calculated using the following pseudocode, with partial charges taken from the Talaris energy function [60].

q1 = partial_charge for atom X of amino acid 1

q2 = partial_charge for atom Y of amino acid 2

absF = 330.0 * abs(q1*q2)/(distance**2)

if q1*q2 < 0.0: absF * = −1.0

if SequenceSeparation ≥ 10: add absF to electrostatic (long-range)

if SequenceSeparation < 10: add absF to electrostatic (short-range)

Final per-residue values were then binned as follows:

bin = np.clip(int(round(residue_value/16.0)), −9, 9)

Cation–Pi. We recalculated the electrostatic scores after adding arbitrary partial charges to the surfaces of aromatic rings, with a partial charge value of −0.05 added 0.85 Å above and below the plane of the ring for each atom, counterbalanced by a partial charge of 0.1 at the atom. The cation–pi score was then taken from the difference between this modified score and the unmodified electrostatic score.

Kinked Beta. Superpositions to kinked beta-fibrils were made for chain A in each of 5 structures, PDB IDs 6bwz, 6bxv, 6bxx, 6bzm, and 6bzp. The full chain of each was superimposed to every overlapping window (same number of residues as the chain with none missing) in our PDB training set, and kinked-beta similarity was measured for each individual PDB residue by taking the minimum CA-RMSD over all of the measurements the residue was involved in. Residues were then classified as K-Beta similar if the minimum CA-RMSD was under 1.0 Å and as K-Beta dissimilar if it was over 2.0 Å.

These 16 physical features were converted to an inferred feature statistics value for every sequence with representation at the residue level and sequence level. At the residue level, each amino acid was represented by 16 × 3 numbers describing the impact of each of the 16 biophysical forces on each residue: (1) the amino acid position number, (2) the score from the comparison to the upper feature value threshold (*W_U_*) and (3) the score from the comparison to the lower feature value threshold (*W_L_*).

Inferred feature statistics for a protein sequence were based on 16 × 20 × *N* matrices, based on three components in the sequence representation, which function as 3 layers of our machine-learning model architecture. (i) A sequence is characterized by 16 physical features acting on each residue. (ii) The impact of each physical feature is dependent on residue type, represented by 20 residue-type groups. (iii) *N* is the number of residues of a specific type within the sequence, with *z* being the position (or index, see below).

Thus, the inferred feature statistical values are determined by translating protein sequences into 16 × 20 × *N* matrices (See Equation (3) and Figure 2).
(3)S=En(seq)[x][y][z]
where
x∈16 features,
y∈20 residues,
z∈N residue positions,
*S—inferred feature statistics value from PDB.*

### 5.6. Predictor Training

Predictor training had the following steps: (1) For each physical feature and each residue type, we set a upper and lower threshold (“weight”) for its inferred feature value, thereby constructing a 16 × 20 × 2 (each feature has two weight values: upper and lower threshold) array (Equation (4)). (2) We initialized the “sum feature score” for each physical score to 0. (3) For each residue in a sequence, if its feature score was higher than the upper threshold, we considered this residue as “abnormally active” in terms of this physical feature, and rewarded the corresponding “sum feature score” by adding 1 to it; if its feature score was lower than the lower threshold, then we considered it as “abnormally inactive” in terms of this physical feature, and penalized the corresponding “sum feature score” by subtracting 1 from it; if its feature score was between the upper and lower thresholds, then we considered it to be “within normal range”, and did nothing (Equations (5) and (6)). (4) By optimizing the AUROC score function (Equation (7)) for each feature, we found the best feature combination and the best weight that maximized the gap of the sum feature score(s) between PS-positive sequences and PS-negative sequences (Equation (8)). (5) By summing “sum feature scores” and training the weights of features using a genetic algorithm, we calculated a “total sum probability” for any sequence, which was the final estimate of its phase-separation ability (Equation (9)).
(4)$$W=Th\left[x\right]\left[y\right]=\left(\begin{array}{c} W_{U}\\ W_{L} \end{array}\right)$$
where
x∈16 features,
y∈20 residues,
*W_U_—upper feature value threshold*,

*W_L_—lower feature value threshold*.

(5)$$f\left(x,seq,W\right)=\overset{20}{\underset{y=1}{\sum }}\overset{N}{\underset{z=1}{\sum }}P\left(En\left(seq\right)\left[x\right]\left[y\right]\left[z\right],W\right)$$
(6)$$P\left(S,W\right)=\sum \left(\left(S>W_{U}\to 1\right)+\left(S<W_{L}\to -1\right)\right)$$
where
x∈16 features,
y∈20 residues,
z∈N residue positions,
*S—Inferred feature values*,

*W—Weights for inferred feature values*.

(7)$$AUC\left(f,X,W\right)=\sum_{x\in X}\frac{\sum_{seq_{1}\in D^{PDB}}\sum_{seq_{2}\in D^{PS}}\left(f\left(x,seq_{1},W\right)<f\left(x,seq_{2},W\right)\to 1\right)}{\left|D^{PDB}\right|\cdot \left|D^{PS}\right|}$$
(8)$$W_{optimum}=\arg\;\underset{X,W}{\max}\;AUC\left(f,X,W\right)$$
where
*X—selected feature combination*,

x∈X features, 
*f—feature score function (Equation (5))*,

$$D^{PDB}—\mathrm{set}\;\mathrm{of}\;\mathrm{sequences}\;\mathrm{from}\;\mathrm{PDB}\;(\mathrm{negative}\;\mathrm{samples}),$$
$$D^{PS}—\mathrm{set}\;\mathrm{of}\;\mathrm{sequences}\;\mathrm{that}\;\mathrm{are}\;\mathrm{PS-positive}\;(\mathrm{positive}\;\mathrm{samples}).$$
(9)$$Pred\left(seq\right)=\underset{x\in X}{\sum }f\left(x,seq,W_{optimum}\right)$$
where
*seq—input sequence*,

*X—selected feature combination*,
x∈X features, 
*f—feature score function (Equation (5))*,
$$W_{optimum}—\mathrm{optimized}\;\mathrm{weights}\;\mathrm{in}\;\mathrm{feature}\;\mathrm{score}\;\mathrm{function}.$$

The optimization process for parameters in a predictive algorithm is called “training”. In this work, the training of the phase-separation predictor had two parts: (i) training of the upper and lower weights of “binary feature score” for 16 features × 20 residue types; (ii) training of the combination of features to include in the final predictor. Numerically, the number of parameters trained was 16 × 20 × 2 weights = 640 weights (16 biophysical forces; 20 residue types; two weights; *W_U_* (upper threshold); *W_L_* (lower threshold)). Another “hyperparameter” being trained here was the selection of biophysical forces to include, with only 8 out of the 16 biophysical forces ultimately being used to avoid overfitting (requiring only 320 weights). The data used for the initial training were from the sequences of the PS-positive proteins in the training set that were separated from the test set (565–260 sequences) and the PS-negative sequences (1703 from either the PDB, human or human + PDB). The data used for training the “final models” included all 565 sequences of the PS-positive proteins and 3406 sequences from either the PDB, human, or PDB + human PS-negative sets.

This training was conducted on the positive and negative training datasets using a genetic algorithm. Specifically, we randomly generated an initial set of 640 weights, and then, for each iteration, we randomly picked a subset of these 640 weights to change and accepted the changes that improved the behavior (loss function based on “genetic operators”). We performed many iterations until a fixed number of generations was reached. The loss function was the AUROC curve (area under the receiver operating characteristic curve) as described above (Equation (7)); the performance of the predictor was then evaluated using the test set as well as by comparison against the baseline models.

Importantly, we used a genetic algorithm to optimize the weights (parameters) with the overall architecture being a 3-layer “neural network”-like predictive model with a non-convex loss function. For more details on implementation of training and prediction, please see https://github.com/julie-forman-kay-lab/LLPhyScore (accessed on 1 July 2022).

### 5.7. Proteome Analysis

Human proteins with scores in the top 10% of the human proteome using 8 predicted single-feature scores as well as the final predictor (8-feature sum score) were separately uploaded to DAVID 6.7 (https://david-d.ncifcrf.gov/ (accessed on 1 July 2022)) [73]. The enrichments of biological process, cellular component, and molecular function GO terms were analyzed for the proteins, with their respective *p*-values (EASE score) obtained. The resulting GO term enrichments were compared against the results in Vernon et al. [29], Vernon et al. [38], and Chu et al. [40].

## Figures and Tables

**Figure 1 biomolecules-12-01131-f001:**
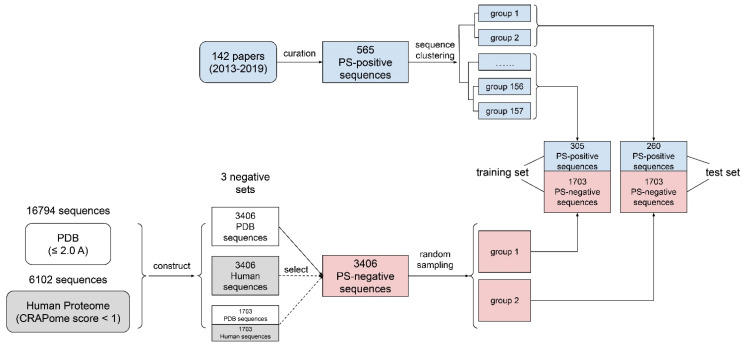
Data curation workflow. A schematic diagram of how data for training were obtained and processed.

**Figure 2 biomolecules-12-01131-f002:**
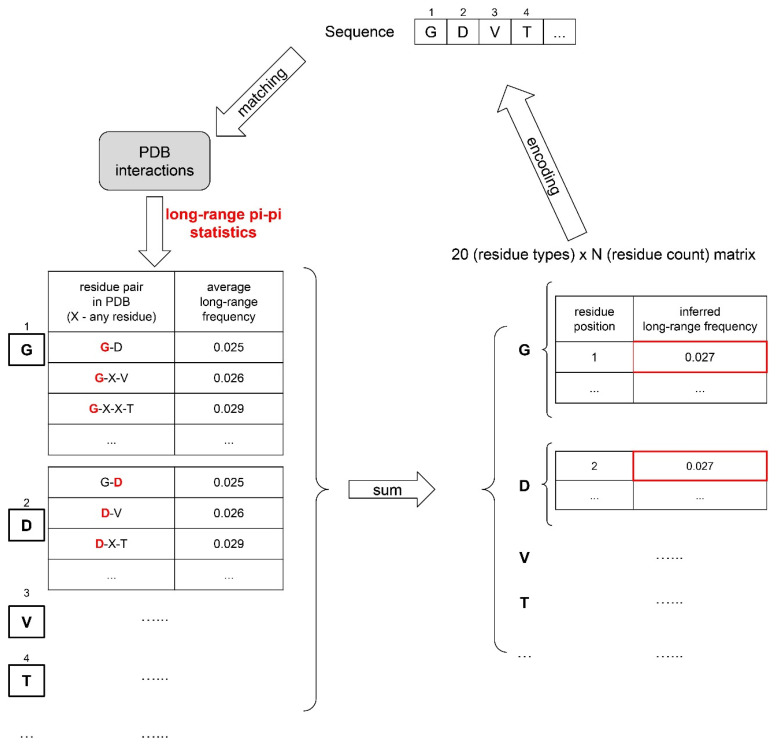
Physical-interaction- and structure-based feature extraction. An example is given of the feature representation of sequences for the sequence “GDVT” converted to the pi–pi (long-range) feature matrix.

**Figure 3 biomolecules-12-01131-f003:**
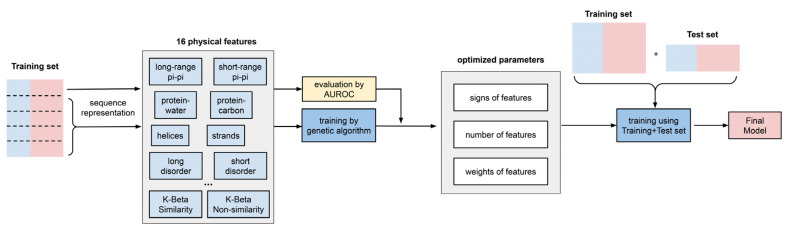
Predictor training workflow. A schematic diagram of the steps in training is shown.

**Figure 4 biomolecules-12-01131-f004:**
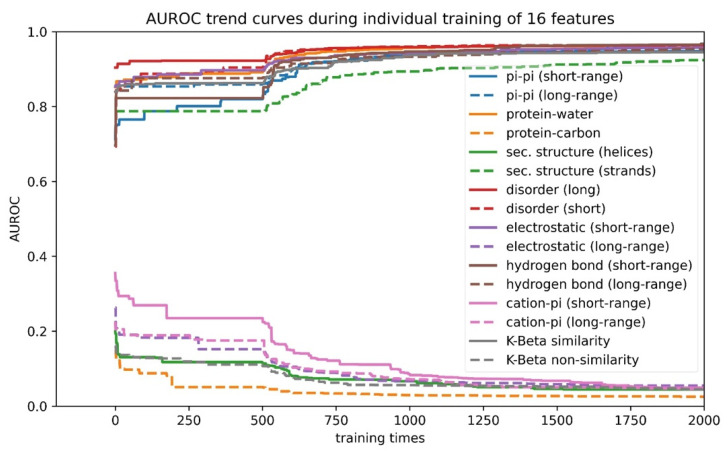
Direction of correlation of features with performance of the developing phase-separation predictor. Training curves of 16 features to reveal the direction of correlation of each feature with score. Features that rise towards AUROC = 1.0 have “positive” features; features that decline towards AUROC = 0.0 have “negative” signs.

**Figure 5 biomolecules-12-01131-f005:**
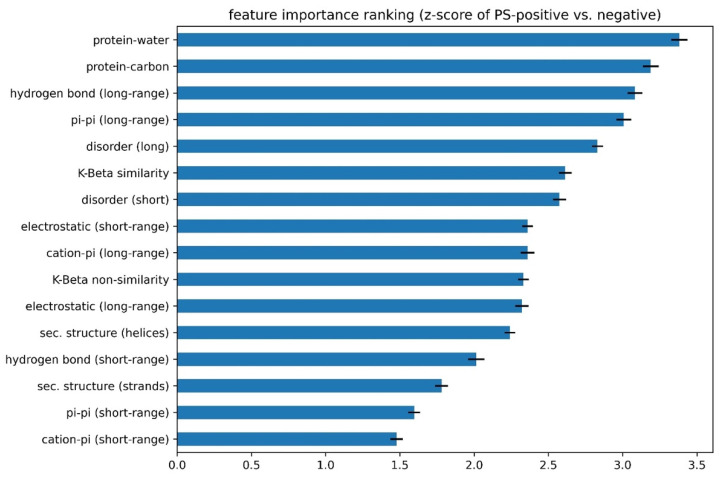
Ranking of the importance of features to discrimination in the developing phase-separation predictor between PS-positive and PS-negative sequences. The z-score of PS-positive sequences’ individual feature values against the mean PS-negative sequences’ values is shown.

**Figure 6 biomolecules-12-01131-f006:**
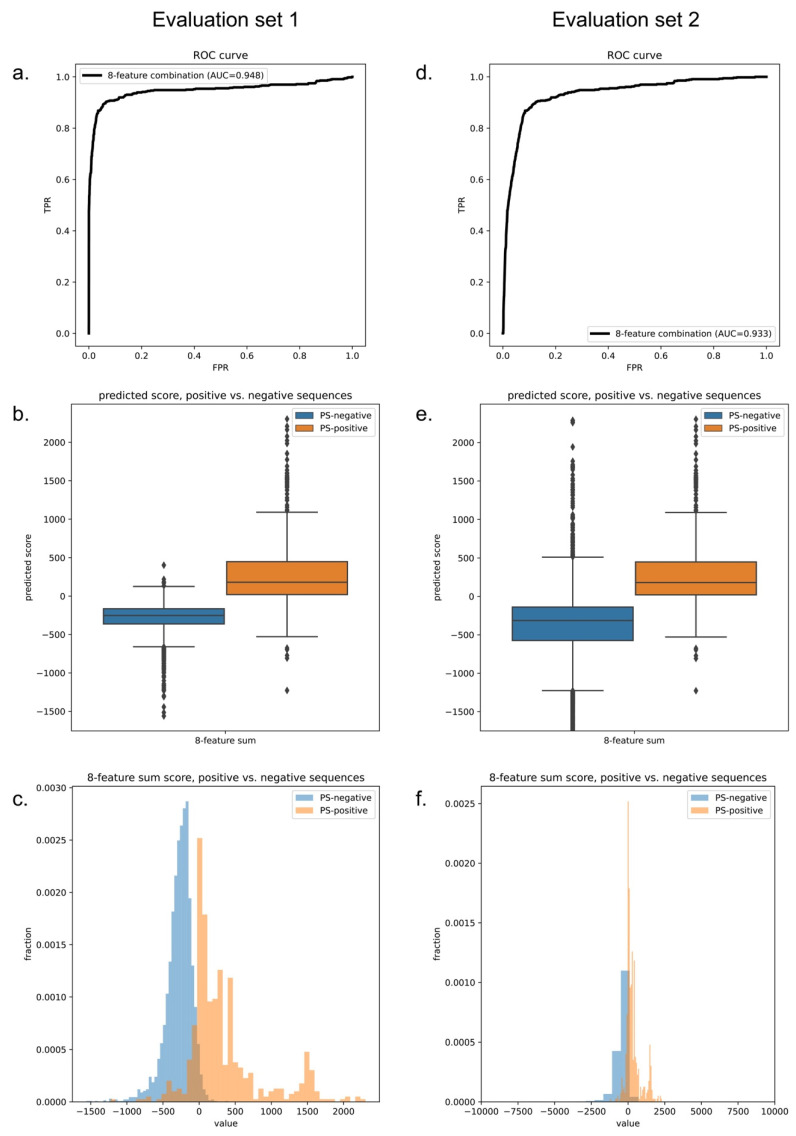
Final predictor of model performance. Performance plots of the final human + PDB model on evaluation set 1 (left, PS-positive sequences and the entire PDB proteome) and evaluation set 2 (right, PS-positive sequences and the entire human proteome). (**a**,**d**) ROC curves. (**b**,**e**) Predicted score boxplots of positive vs. negative sequences. (**c**,**f**) Distribution histograms of positive vs. negative sequences.

**Figure 7 biomolecules-12-01131-f007:**
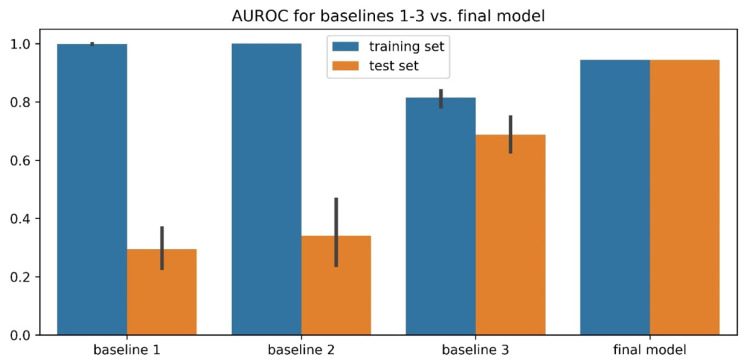
Comparison of three training baselines and the final human + PDB predictor model for validation. Baseline 1 was created by providing random values from a normal distribution N(0, 1) in the weight-training step instead of providing PDB-based physical-feature values to the genetic algorithm. Baseline 2 was created by providing random values from the distribution of residue-specific physical-feature values instead of providing sequence-based physical-feature values. Baseline 3 was created by optimizing 1 weight for 20 residue types for each physical feature (removing residue specificity) during training instead of optimizing 20 weights for 20 residue types for each physical feature.

**Figure 8 biomolecules-12-01131-f008:**
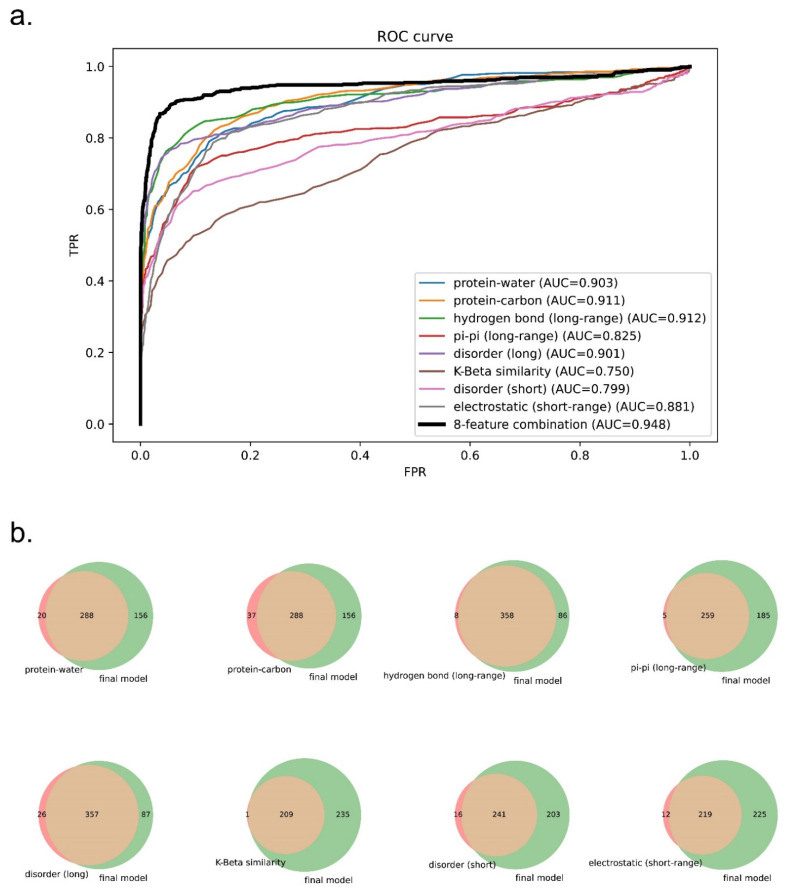
Comparison of the performance of predictors trained on eight features vs. one feature for the human + PDB model. (**a**) ROC curves of one-feature predictors vs. the eight-feature predictor. (**b**) Venn diagrams showing the coverage overlaps of PS-positive sequences by one-feature predictors vs. the eight-feature predictor at a confidence threshold that returns 2% of the PDB.

**Figure 9 biomolecules-12-01131-f009:**
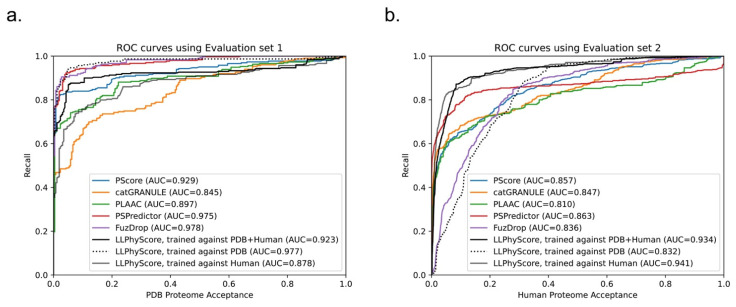
Comparison of LLPhyScore (three models) with other phase-separation predictors. Relationship between percent recall and total percentage of (**a**) evaluation set 1 and (**b**) evaluation set 2 accepted at the given thresholds for PScore, catGRANULE, PLAAC, PSPredictor, FuzDrop and LLPhyScore.

**Figure 10 biomolecules-12-01131-f010:**
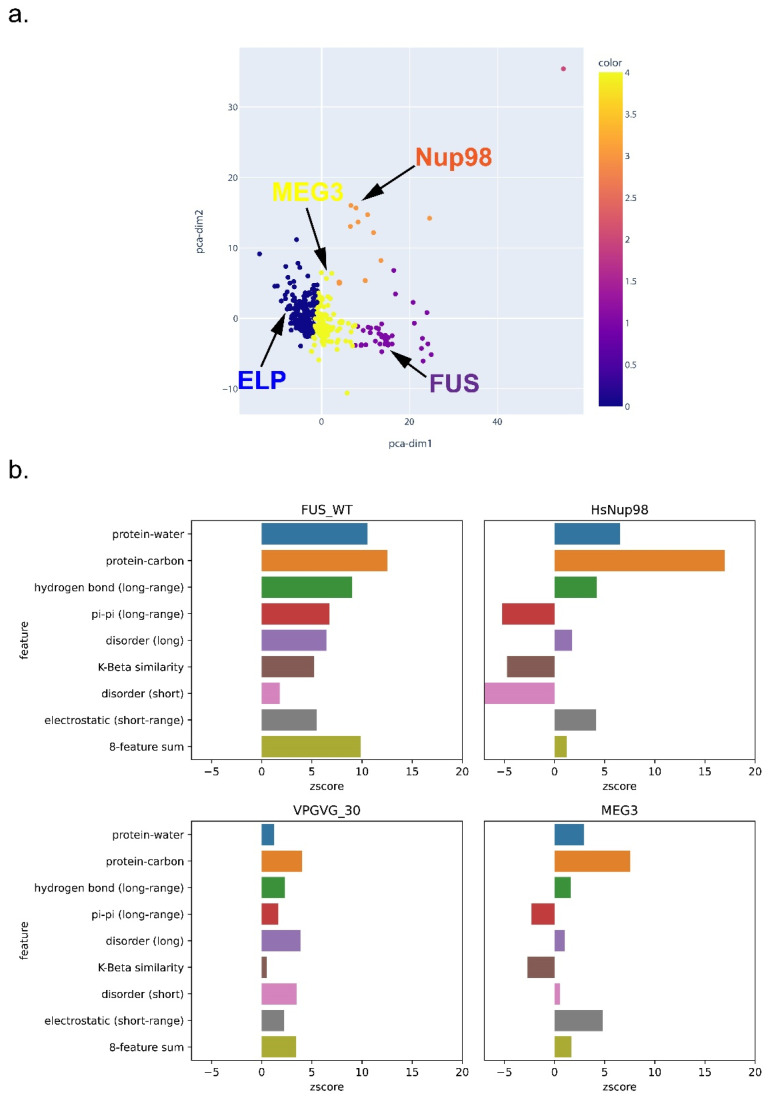
Feature-score-based clustering for PS-positive proteins for the human + PDB model. (**a**) Plot of two abstracted dimensions for clustering based on feature z-scores, showing the separation of different types of phase-separating sequences. (**b**) The score breakdown of four example sequences from four distinct clusters in (**a**): FUS (human), Nup98 (human), elastin-like peptide (ELP, VPGVG_30, 30 repeats of VPGVG) and MEG-3 (*C. elegans*).

**Figure 11 biomolecules-12-01131-f011:**
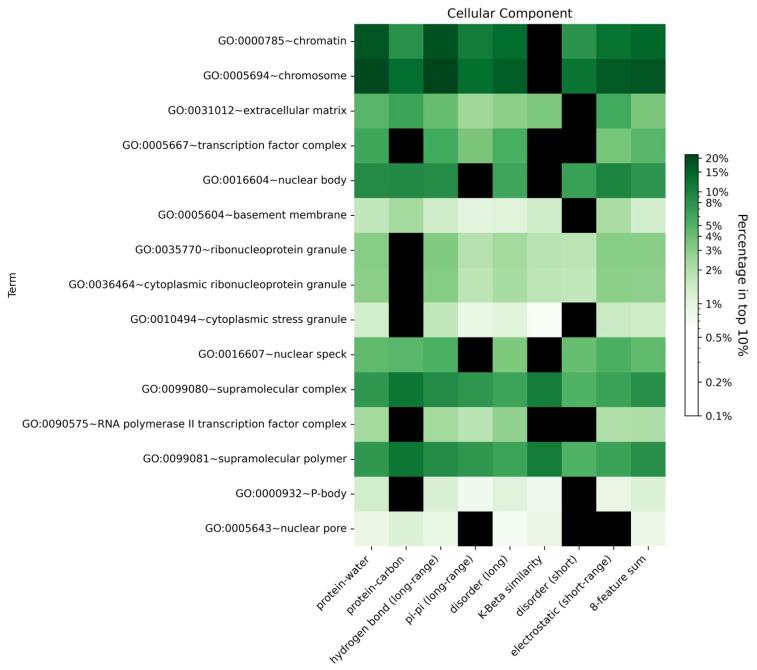
Enrichment heatmap by GO functional annotations for different features for the human + PDB model. Heatmap showing the enrichment of the proteins with a given functional annotation that fall under a 10% confidence threshold for each single-feature score and the eight-feature sum score. The color gradient shows the natural logarithm of the enrichment percentage. The black boxes indicate that no proteins in this GO term are within the top 10% of the corresponding score type.

**Figure 12 biomolecules-12-01131-f012:**
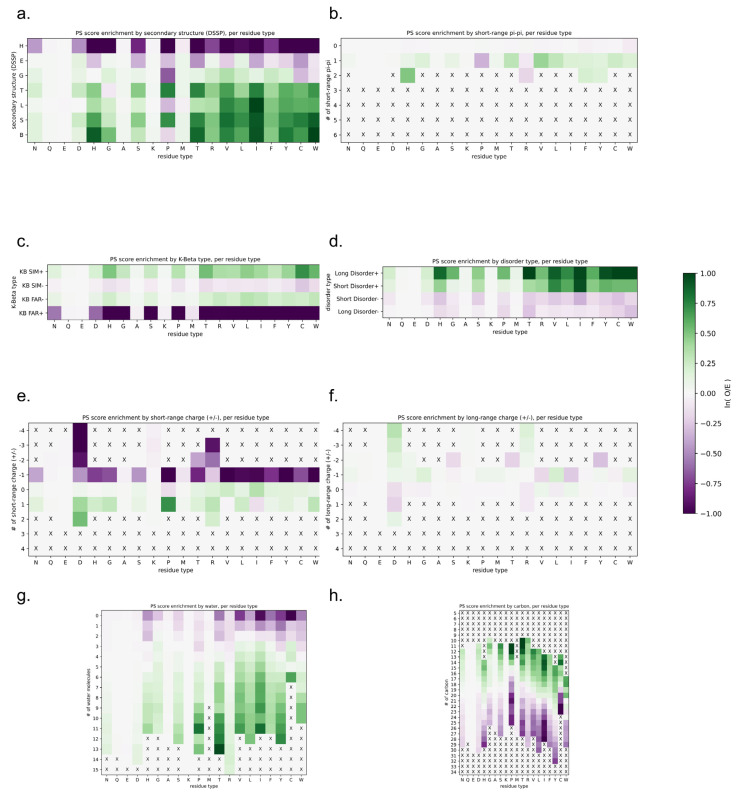
LLPhyScore score enrichment by eight selected physical features for the PDB proteome, per residue type, for the human + PDB model. Heatmaps show the score enrichment in PDB protein sequences by each feature’s discrete values, normalized to each residue type. The color gradient shows the natural logarithm of the observed over expected ratio. Enrichment for (**a**) secondary structure (H, alpha-helix; E, beta-sheet; G, 3_10_ helix; T, hydrogen-bonded turn; L, loop; S, bend; B, single-pair beta-sheet), (**b**) short-range pi–pi, (**c**) K-beta, (**d**) disorder, (**e**) short-range electrostatic, (**f**) long-range electrostatic, (**g**) protein–water and (**h**) protein–carbon. The color bar for all heatmaps is shown at the right.

**Figure 13 biomolecules-12-01131-f013:**
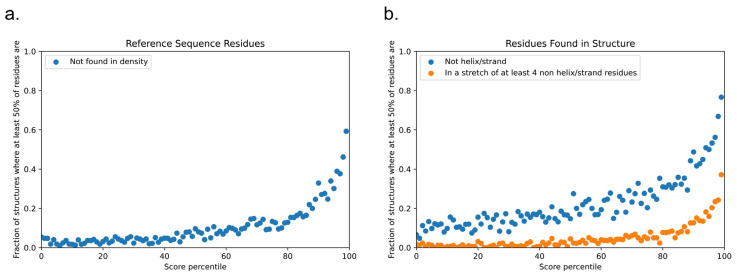
Disordered character of PDB sequences according to the LLPhyScore of chain reference sequences. Panel (**a**) shows the fraction of proteins in each percentile bin of LLPhyScore for which more than 50% of the reference sequence is missing from density (protein sequence that does not show up in the structure). Panel (**b**) shows the disordered/irregular structural character of residues that are within the density in the structure, with blue showing the fraction of proteins in each percentile bin for which more than 50% of the observed residues have a DSSP assignment other than helix or strand, and orange shows the fraction for which more than 50% of such residues are found in stretches of at least four residues in length with no helical or sheet structure.

## Data Availability

Please see https://github.com/julie-forman-kay-lab/LLPhyScore (accessed on 1 July 2022) for details of implementation of training and prediction as well as to utilize the code. Data from analysis regarding functional annotations were separately uploaded to DAVID 6.7 (https://david-d.ncifcrf.gov/ (accessed on 1 July 2022)) [73].

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
