# Peer review of "An Interpretable Machine-Learning Algorithm to Predict Disordered Protein Phase Separation Based on Biophysical Interactions"

_biomolecules, 2022, doi:10.3390/biom12081131_

Round 1

Reviewer 1 Report

This paper reports on a machine-learning algorithm to predict phase separation in protein systems. This algorithm tries to single out the factors that lead to phase separation. As a physical chemist I find this attempt a bit strange in parts. For instances, the authors state that protein-water interactions are important and that earlier authors have overlooked this fact. Of course, it is important and it turns out to be the most important factor indeed as shown in Figure 5. Moreover, the factors enumerated in tables like Figure 5 are not independent from each other. So short-range electrostatics is coupled to long-range electrostatics and not independent. So, Figure 5 tells us that both factors are of the same importance (if I understand it right). Figure 8 at least tries to boil down these factors to 8 most important ones. However, the entire approach ignores all what we already know about the physics of protein-protein interaction. It just correlates experimental findings with structural data without the slightest attempt to weigh factors according to physical importance. In this way it is just treating the entire problem as a black box, as if we would deal with a problem of e.g. botany. At the end, the authors state “Our results are consistent with the understanding that phase separation is driven by a combination of physical factors, including pi-pi contacts, disorder, hydrogen bonding, including in the context of kinked-beta structure, electrostatics, and water interactions.” So, did we learn anything new or non-trivial from this work?

I am afraid that many of these attempts share the same deficiency. However, the authors could at least try to add some more physical insight to this paper.

Author Response

Response to Reviewer Comments

Reviewer 1

Does the introduction provide sufficient background and include all relevant references? Can be improved

Are all the cited references relevant to the research? Can be improved

Is the research design appropriate? Not applicable

Are the methods adequately described? Can be improved

Are the results clearly presented? Must be improved

Are the conclusions supported by the results? Not applicable

Author Reply: We appreciate the reviewer’s comments and have made efforts to improve the clarity of the manuscript throughout, including providing more background and additional references. See the comments below and revised manuscript and supplementary material with highlighted changes (submitted as a supplementary file for review).

Comments and Suggestions for Authors

This paper reports on a machine-learning algorithm to predict phase separation in protein systems. This algorithm tries to single out the factors that lead to phase separation. As a physical chemist I find this attempt a bit strange in parts. For instances, the authors state that protein-water interactions are important and that earlier authors have overlooked this fact. Of course, it is important and it turns out to be the most important factor indeed as shown in Figure 5. Moreover, the factors enumerated in tables like Figure 5 are not independent from each other. So short-range electrostatics is coupled to long-range electrostatics and not independent. So, Figure 5 tells us that both factors are of the same importance (if I understand it right). Figure 8 at least tries to boil down these factors to 8 most important ones. However, the entire approach ignores all what we already know about the physics of protein-protein interaction. It just correlates experimental findings with structural data without the slightest attempt to weigh factors according to physical importance. In this way it is just treating the entire problem as a black box, as if we would deal with a problem of e.g. botany. At the end, the authors state “Our results are consistent with the understanding that phase separation is driven by a combination of physical factors, including pi-pi contacts, disorder, hydrogen bonding, including in the context of kinked-beta structure, electrostatics, and water interactions.” So, did we learn anything new or non-trivial from this work?

I am afraid that many of these attempts share the same deficiency. However, the authors could at least try to add some more physical insight to this paper.

Author Reply: As a physical chemist, the reviewer very well appreciates the importance of protein-water interactions. However, much of the phase separation literature speaks only of protein-protein interactions, ignoring protein-water interactions, and some of it assumes that aromatic interactions are the dominant contributor for all proteins or that other subsets of physical chemical interactions are generally dominant or that particular interaction types are not important (e.g., h-bonding, kinked beta). One goal of our work was therefore to underscore the, perhaps obvious, involvement of solvent and multiple other physiochemical interactions, which we have now emphasized on page 6. Certainly, we agree with the reviewer that many (most) of these interactions are correlated and are not independent. We have made this point more clear in the manuscript in multiple places, including the abstract, bottom of page 3, top of page 8, bottom of page 9, top of page 16, page 20, and the conclusion on page 24. 

            The reviewer also highlighted our deficiency in more clearly stating our initial hypothesis with this work and what we learned. We did initially assume that proteins found in distinct biomolecular condensates would use specific types of physical “interactions” based on our definitions as a way to generate specific condensates, now explicitly stated on page 6. What we found is that all of these physical interactions are highly interdependent and most proteins use combinations of them all. Thus, there are few clear distinctions, stated on page 16 and in the conclusion. We also found that almost all the terms are correlated to solvation so that the idea of protein-protein interactions itself is simplistic and likely for solvated condensates there is always a three-way interaction involving 2 protein groups and water, also included now in the conclusion (page 24).

Reviewer 2 Report

In the paper by Forman-Kay and coworkers, the authors describe their development of a sequence-based predictor (LLPhyScore) for identifying proteins that exhibit LLPS behavior that also is able to predict the underlying mechanism(s) driving the phase separation. Besides the predictor, the authors provide highly curated datasets of negative and positive control for LLPS, which should be very useful to the field. The methodology to develop LLPhyScore and curate the datasets were both reasonable and described well. The comparison of LLPhyScore results to the results from other predictors (catGranule, PLAAC, PSPredictor, FuzDrop) gives confidence the new predictor works as described. I fully support publishing this work. I have a few comments that the authors may wish to reply.

1. The authors curated the human proteome by applying CRAPome scores, removing ~70% of the sequences. This curated set was used as one of the negative control sets. For the removed sequences, does LLPhyScore (and other predictors) predict a high percentage as LLPS drivers, justifying their removal? Did this curation overall deplete or enrich the human proteome for phase-separating sequences? I would guess depletion, but with many predictors showing ≤10% of the human proteome as phase-separating, it's hard to judge the 70% removal rate. They state, "Certainly, there are fewer positives in these sets than in the known PS-positive sequences, ...". It would be helpful here to provide quantitative measures. For example, what is the AUROC for the original uncurated (as the test set) compared to curated proteome (as the comparison set)? It should be >0.5 if curation worked correctly, no? Their tests of comparing results from differents mixtures of the negative sets showed consistency, but not whether curation was as intended.

2. The negative correlation of LLPS and helix (Table S3, Figure 12), does that result support the idea that helix formation inhibits LLPS? This is discussed somewhat on pg 20 with "Fully self-satisfied structures, specifically helices, have the lowest scores." etc. But it is not entirely clear to me if extrapolations to structural hypotheses can be made, i.e., from redundancy owing to correlated trends. Following this line, negative correlations to long- and short-range cation-pi interactions are found (Table 3). Does this result contradict experiment (e.g., Wang et al Cell 2018)? A broader discussion of the cation-pi result is likely warranted but was absent.

3. It would be interesting to compare proteins LLPhysScore identifies as phase-separating that PScore does not. And vice versa. Figures 9a and b suggests there are some based on comparing AUC values. Would this be owing to cation-pi, because these interactions were omitted from the 8 features of LLPhyScore? So, removing only redundancy with the 16 to 8 feature change wasn't necessarily achieved?

4. Figure S1 is unreadable.

5. A key for the color gradient is missing in Figures 11 and S11. Should we assume red/black/blue = high/med/low enrichment? Does collagen have no enrichment for any of the molecular features utilized by LLPhyScore (if I'm interpreting Fig 11 correctly)?

Author Response

Response to Reviewer Comments

Reviewer 2

Does the introduction provide sufficient background and include all relevant references? Yes

Are all the cited references relevant to the research? Yes

Is the research design appropriate? Yes

Are the methods adequately described? Yes

Are the results clearly presented? Can be improved

Are the conclusions supported by the results? Yes

Author Reply: We appreciate the reviewer’s positive comments and have also worked to increase the clarity of our manuscript in multiple places.

Comments and Suggestions for Authors

In the paper by Forman-Kay and coworkers, the authors describe their development of a sequence-based predictor (LLPhyScore) for identifying proteins that exhibit LLPS behavior that also is able to predict the underlying mechanism(s) driving the phase separation. Besides the predictor, the authors provide highly curated datasets of negative and positive control for LLPS, which should be very useful to the field. The methodology to develop LLPhyScore and curate the datasets were both reasonable and described well. The comparison of LLPhyScore results to the results from other predictors (catGranule, PLAAC, PSPredictor, FuzDrop) gives confidence the new predictor works as described. I fully support publishing this work. I have a few comments that the authors may wish to reply.

Author Reply: We are pleased that the reviewer finds our work to be a valuable contribution.

  1. The authors curated the human proteome by applying CRAPome scores, removing ~70% of the sequences. This curated set was used as one of the negative control sets. For the removed sequences, does LLPhyScore (and other predictors) predict a high percentage as LLPS drivers, justifying their removal? Did this curation overall deplete or enrich the human proteome for phase-separating sequences? I would guess depletion, but with many predictors showing ≤10% of the human proteome as phase-separating, it's hard to judge the 70% removal rate. They state, "Certainly, there are fewer positives in these sets than in the known PS-positive sequences, ...". It would be helpful here to provide quantitative measures. For example, what is the AUROC for the original uncurated (as the test set) compared to curated proteome (as the comparison set)? It should be >0.5 if curation worked correctly, no? Their tests of comparing results from differents mixtures of the negative sets showed consistency, but not whether curation was as intended.

Author Reply: The reviewer raises an important point regarding comparison of curated and noncurated human proteome sets. We realize that we were not always clear regarding which of these was used for presentation of results and we have now clarified this in multiple instances, pages 11, 12 and 15, as well as remaking figures. We use the curated set for training but plot the AUROC and other data against the full human proteome (uncurated) plus positive sequences (Evaluation set 2), as our predictor should certainly perform better against the curated set based on this. To clearly demonstrate the difference, we now include a new Supplementary Figure S12 with histograms of the LLPhyScore values for these two sets, showing that the curated set is clearly shifted to lower values. We performed a Z-test to quantify the statistical significance of the difference between to the scores for these two sets and found a highly significant P-value of 5E-60. The LLPhyScore and CRAPome score values for all human sequences are provided in a new Supplementary Table S3. Clear discussion of these points is now included in the text on pages 5 and 25.

  1. The negative correlation of LLPS and helix (Table S3, Figure 12), does that result support the idea that helix formation inhibits LLPS? This is discussed somewhat on pg 20 with "Fully self-satisfied structures, specifically helices, have the lowest scores." etc. But it is not entirely clear to me if extrapolations to structural hypotheses can be made, i.e., from redundancy owing to correlated trends. Following this line, negative correlations to long- and short-range cation-pi interactions are found (Table 3). Does this result contradict experiment (e.g., Wang et al Cell 2018)? A broader discussion of the cation-pi result is likely warranted but was absent.

Author Reply: We very much appreciate the reviewer’s comment since we did not make it clear that the signs of the individual physical interaction terms within our predictor cannot be interpreted simply as positive or negative correlation with phase separation. Importantly, the contribution to the score and the contribution to phase separation are not simply related, even if the distinction between protein-water and protein-carbon interactions is suggestive of such a simplistic interpretation. This is partly due to the lack of independence of the various terms based on our definitions and partly due to the issue with electrostatics having attractive and repulsive contributions based on opposite and same charges, respectively. One example of the interdependence of terms is for cation-pi, which could be seen in some sense as a combination of amino acid preference, electrostatics and pi-pi. Thus, cation-pi is represented but we were not able to represent it discretely as its own term. It is also important to note that the distinction between the 8 top terms and the 8 bottom terms is not dramatic and we used the top 8 to minimize the number of fitted parameters, rather than as a statement that those physical interactions are not important for phase separation. We now clarify this on pages 9 and 10. Greater interpretability comes later based on normalizing by residue type, partly shown with the individual components of the score shown for different proteins (Figure 10), and discussed in more detail on pages 20-23 and Figure 12.

  1. It would be interesting to compare proteins LLPhysScore identifies as phase-separating that PScore does not. And vice versa. Figures 9a and b suggests there are some based on comparing AUC values. Would this be owing to cation-pi, because these interactions were omitted from the 8 features of LLPhyScore? So, removing only redundancy with the 16 to 8 feature change wasn't necessarily achieved?

Author Reply: While the reviewer asks an interesting question, unfortunately a comparison with PScore is not really valuable since the phase-separation-positive set has increased dramatically since PScore was trained. Currently PScore predicts as negative many of the proteins that are in our current positive set. In addition, the behavior of a predictor that is trained with a limited set of interactions cannot be assumed to report only on phase separation by that set of interactions, since it is trying to optimize the recall statistics for the full positive set. Again, LLPhyScore does not remove cation-pi, it merely represents cation-pi by a combination of other terms, and we comment on this point on pages 9 and 10.

  1. Figure S1 is unreadable.

Author Reply: We agree that the data previously represented in Figure S1 should not be shown as a figure within the context of an 8 1/2” x 11” document. Instead, we now have these data as Supplementary File S2 that can be visualized more easily and have removed it as a figure, renumbering the other supplementary figures.

  1. A key for the color gradient is missing in Figures 11 and S11. Should we assume red/black/blue = high/med/low enrichment? Does collagen have no enrichment for any of the molecular features utilized by LLPhyScore (if I'm interpreting Fig 11 correctly)?

Author Reply: We appreciate the reviewer pointing this out. We have changed the colors to be more intuitive and now include a color gradient, a label, and a definition of enrichment in the legend. The low enrichment of score terms for collagen is likely due to the fact that, while some collagens have IDRs that are critical for function, many do not so there is weak enrichment overall. We removed the collagen GO terms and instead have included a smaller selected list of GO terms for biomolecular condensates that are more commonly understood to have contributions from IDR phase separation. We have also increased from 2% to 10% to include more proteins in the list, based on understanding that at least 30% of the proteome is likely involved in condensates. We now have the analysis and figures for all three models in Figures 11, S10 and S11, and have included the full GO term analysis in Supplementary Tables 8A, 8B, and 8C.   

Reviewer 3 Report

The manuscript by Cai et al. reports on a machine learning tool (the hype of the moment, just missing a little bit of alphafold and cryo-EM) to predict LLPS. I start by stating clearly that I am NOT an expert in the LLPS and/or IDPs and I am very skeptical about whatever is collectively called "machine learning". However, I appreciate the effort the authors have put into stating the physical significance of their model. 

On these grounds, I will only state the things that I do not find clear in this manuscript.

1) The negative sequences. These are obtained from the PDB, which in turn is mainly coming from XRD. Crystals for XRD are often grown under conditions that induce LLPS by adding the so-called "crystallization reagents" (10.1038/nprot.2007.198). Even if many proteins would not undergo LLPS without the reagents, I wonder if this could represent a bias. A quick workaround would be to use NMR and Cryo-EM structures only. While this would narrow down the number of unique sequences significantly (85% of the structures are from X-ray), I think this could help, and thus flag it as a major revision needed.

2) Evaluation set 2. I might be missing something, but figure 6, panel e shows that there is no statistical discrimination between PS-positive and PS-negative sequences. Why is this so? It should be explained in simpler terms wrt the present manuscript. Panel b is also taletelling: the outliers are relatively few above the negative set and below the positive set. What features do these structures have? Perhaps Evaluation set 2 should be reconsidered.

Author Response

Response to Reviewer Comments

Reviewer 3

Does the introduction provide sufficient background and include all relevant references? Yes

Are all the cited references relevant to the research? Yes

Is the research design appropriate? Can be improved

Are the methods adequately described? Yes

Are the results clearly presented? Yes

Are the conclusions supported by the results? Can be improved

Author Reply: We appreciate the reviewer’s highlighting of potential for improvement in research design and conclusions and address these below.

Comments and Suggestions for Authors

The manuscript by Cai et al. reports on a machine learning tool (the hype of the moment, just missing a little bit of alphafold and cryo-EM) to predict LLPS. I start by stating clearly that I am NOT an expert in the LLPS and/or IDPs and I am very skeptical about whatever is collectively called "machine learning". However, I appreciate the effort the authors have put into stating the physical significance of their model. 

Author Reply: Machine learning is just a way to describe advanced computational modeling approaches driven by data, such as found in the PDB. Some machine learning approaches, such as PSPredictor, do not enable interpretability, but other designs can. We appreciate the reviewer’s acknowledgement of our focus on physical interpretation of our computational models.   

On these grounds, I will only state the things that I do not find clear in this manuscript.

1) The negative sequences. These are obtained from the PDB, which in turn is mainly coming from XRD. Crystals for XRD are often grown under conditions that induce LLPS by adding the so-called "crystallization reagents" (10.1038/nprot.2007.198). Even if many proteins would not undergo LLPS without the reagents, I wonder if this could represent a bias. A quick workaround would be to use NMR and Cryo-EM structures only. While this would narrow down the number of unique sequences significantly (85% of the structures are from X-ray), I think this could help, and thus flag it as a major revision needed.

Author Reply: We completely agree with the reviewer that the PDB is not a good negative dataset and spend a significant part of our manuscript discussing this and describing an alternative negative dataset based on curation of the human proteome. This leads to three separate models with the PDB, the curated human proteome and a combination of these as the negative sets for training. We also analyze the high-scoring sequences of the PDB (page 22) and show that these have significant disorder relative to the average structure found within the PDB (Figure 13). We reiterate this point now on page 3.

            The suggestion to use sequences from NMR and cryoEM structures could lead to a lower false negative rate but these sequences are significantly biased. For NMR, the structures determined are mostly of relatively short sequences and, since longer IDRs have a much greater propensity to phase separate, the use of NMR sequences as negative will not be helpful. Using cryoEM sequences may be better but they are highly biased to specific classes of proteins, such as membrane proteins. Recognizing that the structures from X-ray crystallography represent more than 90% of PDB and that many other phase-separation prediction tools have used the PDB as a negative set, we chose to include the PDB as a negative set but to focus on our curated human proteome negative set or using that in combination with the PDB. Based on these arguments, we strongly feel that redoing our work with sequences from NMR and cryoEM structures as negative sets would not be valuable. In addition to adding more clarifying text to the manuscript (page 3), we have included the reference (10.1038/nprot.2007.198) suggested by the reviewer.

2) Evaluation set 2. I might be missing something, but figure 6, panel e shows that there is no statistical discrimination between PS-positive and PS-negative sequences. Why is this so? It should be explained in simpler terms wrt the present manuscript. Panel b is also taletelling: the outliers are relatively few above the negative set and below the positive set. What features do these structures have? Perhaps Evaluation set 2 should be reconsidered.

Author Reply: We appreciate the reviewer pointing out that Figure 6 panel e was plotted poorly to confuse readers regarding the discrimination between PS-positive and PS-negative sequences. Our original plot had a y-axis defined to include all outliers. We have now replotted Figure 6 panel e to expand the y-axis scale, making it obvious that the sets are significantly different with the bulk of scores not overlapping (see figure panel below). We have also calculated both T- and Z-tests and found such low P-values we could not discriminate them from 0 (i.e., <<1E-12), representing very strong statistical significance for the distinction between the sets. Based on this we do not agree that Evaluation set 2 (all human proteome sequences) should be reconsidered for evaluation, but emphasize that we use a curated subset of the human proteome for training.  

            For Figure 6 panel b using Evaluation set 1 (PS-positive set plus all PDB sequences), we have examined the outliers above the negative set and below the positive set, as suggested. We note that panel b is for Evaluation set 1 so insights regarding these outliers do not speak to Evaluation set 2, but Evaluation set 1. For the sequences below the positive set, these are expected since the predictor does not have an AUROC of 1. The exact reasons for these sequences being poorly predicted are challenging to understand due to the complexity of the training, including residue preference, numbers, positions and the 8-features, as well as the combined Human+PDB negative training set. The outliers above the negative set would likely be explained by the analysis of the highest scoring PDB sequences which we do on page 23 and in Figure 13. We found that these sequences are from proteins with significant disorder relative to the average structure in the PDB, both in terms of missing density and for residues within density being enriched in irregular or loop structure, reinforcing the idea that the PDB is not a good negative set, a view that is emphasized in the text here and elsewhere.

Round 2

Reviewer 3 Report

My comments have been addressed only in part, and my concerns remain unchanged, so does my recommendation. 

Author Response

We appreciate the reviewer's continued concern with sequences from PDB X-ray crystal structures as a negative training set. In addition to our previous comments, we would like to provide more insight into the process of developing a predictive model to assist in understanding our choices. It is worth noting that the accuracy of the negative/positive labels is only one consideration when training a predictive model. When generating our negative sets, we are equally as concerned with the distribution of physical and other properties, a bar that sequences of NMR and cryoEM structures does not meet. 

(a) If we have a sequence length imbalance between the positive and negative sets then the training task becomes biased towards a trivial solution that does not transfer to performance on other tasks. NMR structures as a rule are short sequences, typically a single domain under 100 residues, and as such these sequences are far smaller than the vast majority of natural human proteins. Telling them apart from known phase separating proteins is, for the most part, trivial. 

(b) Similarly, cryoEM structures have a significant bias towards membrane proteins, and our expectation is that biasing the negative set in that way would cause the predictor to learn merely that phase separating proteins do not have transmembrane domains.

This is a complicated problem and there are no simple solutions. By showing three different curated negative sets, which we understand to reflect a broader distribution of sequence properties than sequences from NMR or cryoEM structures, we are intending to cast new light on the scope of the problem. We also present a solution that, while not perfect, reflects an advance over prior work. 

Ultimately, the field needs an empirical negative set, from proteins that have been tested in a consistent and rigorous fashion, matching the treatment of the positive set. This type of dataset is not trivial to prepare, but we hope the work we present here can be of use in stimulating and planning for its collection.